# Detection and Verification of QTL for Salinity Tolerance at Germination and Seedling Stages Using Wild Barley Introgression Lines

**DOI:** 10.3390/plants10112246

**Published:** 2021-10-21

**Authors:** Mohammed Abdelaziz Sayed, Rasha Tarawneh, Helmy Mohamed Youssef, Klaus Pillen, Andreas Börner

**Affiliations:** 1Agronomy Department, Faculty of Agriculture, Assiut University, Assiut 71526, Egypt; 2Resources Genetics and Reproduction, Gene Bank, Leibniz Institute of Plant Genetics and Crop Plant Research (IPK), OT Gatersleben, D-06466 Seeland, Germany; tarawneh@ipk-gatersleben.de; 3Faculty of Agriculture, Cairo University, Giza 12613, Egypt; helmy.mohamed-youssef-ibrahim@landw.uni-halle.de; 4Plant Breeding, Institute of Agricultural and Nutritional Sciences, Martin-Luther-University Halle-Wittenberg, Betty-Heimann-Str. 3, 06120 Halle, Germany; klaus.pillen@landw.uni-halle.de

**Keywords:** *Hordeum spontaneum*, salt-tolerance, early stages, introgression lines, quantitative trait loci (QTL)

## Abstract

Salinity is one of the major environmental factors that negatively affect crop development, particularly at the early growth stage of a plant and consequently the final yield. Therefore, a set of 50 wild barley (*Hordeum vulgare* ssp. *spontaneum*, *Hsp*) introgression lines (ILs) was used to detect QTL alleles improving germination and seedling growth under control, 75 mM, and 150 mM NaCl conditions. Large variation was observed for germination and seedling growth related traits that were highly heritable under salinity stress. In addition, highly significant differences were obtained for five salinity tolerance indices and between treatments as well. A total of 90 and 35 significant QTL were identified for ten investigated traits and for tolerance indices, respectively. The *Hsp* introgression alleles are involved in improving salinity tolerance at forty (43.9%) out of 90 QTL including introgression lines S42IL-109 (2H), S42IL-116 (4H), S42IL-132 (6H), S42IL-133 (7H), S42IL-148 (6H), and S42IL-176 (5H). Interestingly, seven exotic QTL alleles were successfully validated in the wild barley ILs including S42IL-127 (5H), 139 (7H), 125 (5H), 117 (4H), 118 (4H), 121 (4H), and 137 (7H). We conclude that the barley introgression lines contain numerous germination and seedling growth-improving novel QTL alleles, which are effective under salinity conditions.

## 1. Introduction

Salinity, together with drought, is a serious constraint to food security in many parts of the world due to it suppressing plant growth, development, and crop productivity, and restricting the use of agricultural land. However, about 20% of total cultivated and 33% of irrigated agricultural lands are afflicted by high salt stress, and the salinity problem is predicted to be increased annually by 10% due to various reasons, including poor irrigation management and climatic change, which causes low precipitation and high surface evaporation [1,2,3], especially as 92% of the salt-affected areas are located in the regions with arid climate [4]. Pirasteh-Anosheh et al. [5] pointed out that plants are exposed to four different types of salinity effects under salt conditions: briefly (1) soil salinity reduces water uptake of a plant due to low soil water potential, which interferes with the osmotic gradient [6]; (2) salinity accelerates the production of active oxygen radicles, like hydrogen peroxide, superoxide, singlet oxygen, and hydroxyl radicle, which may damage or kill plants [7]; (3) the salt interacts with minerals, causing nutrient imbalance and deficiency [8]; and (4) the absorbed salt reaches a level that causes severe cellular toxicity due to low sequestration of Na^+^ into vacuoles [8].

Barley, the fourth most leading cereal crop worldwide, is one of the main winter cereals in the Mediterranean region [9]. Barley is relatively salt-tolerant and has the advantage of growing in marginal environments that are unsuitable for other cereal crops [9,10]. Therefore, it is oftentimes used as a model plant for understanding salinity adaptation mechanisms in crops and for studying the germination in monocots [11,12]. However, salinity causes a significant reduction in its growth and grain yield. The adaptation of barley to salt stress differs from one growth stage to another, where the germination and the early development stages of plant growth are considered the most sensitive ones. Many studies investigated the effect of salt stress on barley seed germination and seedling development to determine the effect of salinity on plants and develop salt-tolerant accessions for breeding programs [10,13,14]. A clear reduction in germination percentage, rate, shoot length, root length, root fresh and dry weights, and relative growth rate was reported by increasing salinity levels in barley [10,13,14,15,16,17].

During the current climate change conditions, crop improvement efforts to abiotic stresses face a great challenge. Therefore, a deeper understanding of crop genetic resources is needed, if these assets contribute effectively in developing improved varieties [18]. Wild barley (*Hordeum vulgare* ssp. *spontaneum* Koch, *Hsp*) has developed unique mechanisms for surviving harsh environments, mainly through forming new genetic variations and alleles [19]. Among various mapping strategies, introgression lines (ILs) breeding is an effective method for improving barley and expanding genetic diversity to meet present and future challenges to crop production [18]. The ILs produced by advanced backcrossing carry only a small introgression of a wild donor parent and represent the complete genome of a wild donor species, which could be utilized as an immortal QTL mapping population [20]. Additionally, *Hsp* has a great potentiality in barley improvement as a donor in the introgression libraries, since positional cloning of natural QTL plays a prevailing role in elucidating the molecular control of abiotic stresses tolerance such as drought and salinity [19]. In this way, by using genetic approaches, new genes and/or new alleles associated with enhanced salinity tolerance at known loci present in wild genotypes could be mapped and targeted for introgression. In the past decade, a number of studies have investigated the S42ILs library under different environmental conditions including drought, phosphorus and nitrogen deficiency [21,22,23,24,25,26]. These experiments identified beneficial exotic QTL alleles, which were contributed significantly to the enhancement of agronomic, biotic, and abiotic stress tolerance-related traits. Further, they concluded that beneficial wild barley QTL alleles are present in the S42ILs library, which could be used to select for improved interesting traits in barley breeding. The present work is considered the first QTL study, conducted under control and salinity stress by applying different NaCl concentrations (75 and 150 mM) at germination and seedling stages using a set of wild barley ILs. Our study aimed at (1) detecting the QTL in the S42IL library associated with salinity tolerance at germination and seedling stages, (2) finding the desirable *Hsp* alleles that improve trait performances under salinity conditions, and (3) verifying and comparing the obtained QTL with previously mapped QTL of a field trial conducted in Egypt under severe salinity conditions using S42 barley population.

## 2. Results

### 2.1. Variations and Heritability

The analysis of variance revealed highly significant differences between genotypes tested for all measured traits under control and salinity conditions at germination and seedling stages (Table 1). Additionally, the combined ANOVA revealed that all effects of genotypes, treatments, and genotype by treatment interactions effects were highly significant for all studied traits. Further, in most cases, coefficients of variations increased by increasing salt concentrations. The genotypes exhibited significant differences in seed viability tested before experiment initiation. As an average of all genotypes, the seed viability was 85%, ranging between 65 and 100%. In addition, moderate to high values of coefficient of determination (R^2^) and heritability (H_b_) estimates were obtained under control and salinity stress conditions. For instance, the highest R^2^ and H_b_ estimates were found for GI, ShL, and SL under control, 75 mM NaCl, and 150 mM NaCl treatments, respectively. Whereas RSR, MGT, and WCP revealed the lowest estimates of R^2^ and H_b_ under the previous conditions, respectively.

### 2.2. Trait Means and Reduction Percentage

Trait means, and ranges under control and salinity conditions as well as reduction percentage (R%) due to salinity compared to control are presented in Table 2. Most of the investigated traits were declined by increasing salt concentrations compared to control, except RSR and SDW were increased under 150 mM NaCl treatment by 54.5 and 23.4%, respectively, compared to control. For example, as an average overall genotypes, the means values of GP (Figure 1A) were 89.9, 80.3, and 79.6%; ShL means were 9.5, 6, and 4.1 cm; RL means (Figure 1F) were 10.6, 8, and 6.2 cm; and WCP means (Figure 1K) were 89, 87.9, and 79% under control, 75 mM NaCl, and 150 mM NaCl treatments, respectively. In addition, maximum R% under 75 mM NaCl treatment was observed for SVI (−37.1%) followed by ShL (−36.8%), while WCP (−2.1%) exhibited the minimum R%. Similarly, under 150 mM NaCl treatment, the maximum R% were obtained from ShL (−56.8%) followed by SVI (−55.4%), whereas minimum R% was recorded for MGT (−3.2%).

As an average under control, Scarlett showed significant superiority in GP (93.3%), MGT (2.6), SVI (1986), ShL (9.9 cm), RL (11.3 cm), SL (21.3 cm), SFW (278.7 mg), and WCP (90.3%) compared to the S42Ils set, which accounted 89.7%, 3.1, 1795, 9.4 cm, 10.6 cm, 20 cm, 252.5 mg, and 89% for the same traits, respectively. Whereas, under 150 mM NaCl conditions, the S42IL set had better values in GP (79.6%), GI (59.3), MGT (3 days), SVI (799.8), RL (6 cm), RSR (1.7), and SDW (34.4 mg) compared to Scarlett (Appendix A). Many S42ILs showed remarkable GP and SVI under control and salinity conditions compared to Scarlett, like S42IL-109, S42IL-112, S42IL-116, S42IL-117, S42IL-122, and S42IL-128. Similarly, lines S42IL-106, S42Il-111, S42IL-118, and S42IL-122 had higher GI and better MGT compared to Scarlett under the higher level of salinity. Interestingly, the ILs S42IL-109, S42IL-132, and S42IL-176 exhibited remarkable superiority in ShL, RL, and SL under both salinity treatments compared to Scarlett. They recorded (9.6, 6.6, and 7.2 cm) and (7.7, 7.6, and 7 cm) for ShL, (9.7, 8.2, and 9.9 cm) and (7.3, 7.4, and 7.5) for RL, and (19.2, 14.8, and 17 cm) and (15, 15.1, and 14.6 cm) for SL under 75 mM NaCl and 150 mM NaCl treatments, respectively. Several ILs showed remarkable values of RSR under salinity treatments compared to Scarlett, for instance, S42 IL-125 and S42 IL-127 registered 2.6 and 4.1 as the highest RSR under 75 and 150 mM NaCl treatments, respectively. Under severe salinity conditions, the ILs S42 IL-132, S42 IL-109, and S42 IL-120 gave the highest SFW and recorded 241.8, 224.7, and 217.2 mg compared to Scarlett, which gave 172 mg. For SDW, the ILs S42 IL-132, S42 IL-126, and S42 IL-120 were the best in SDW and gave 45.5, 43.3, and 42.6 mg compared to Scarlett, which recorded 33.5 mg. Additionally, the ILs S42IL-109 and S42 IL-148 had higher WCP by 92.1 and 83.9% under 75 and 150 mM NaCl treatments, respectively.

### 2.3. Phenotypic Correlation among Studied Traits

Pearson correlation coefficients were calculated among the studied traits only under control and 150 mM NaCl treatments, and are displayed in Figure 2. Under both treatments, GP was positively and highly significantly correlated with GI and SVI. ShL showed highly significant and positive correlations with SVI (0.77 and 0.88), RL (0.82 and 0.83), SL (0.96 and 0.97), SFW (0.65 and 0.89), and WCP (0.44 and 0.89) under control and 150 mM NaCl treatments, respectively. Additionally, RL exhibited significant and positive correlations with SL (0.95), SFW (0.63), and WCP (0.41) under control treatment. Whereas under 150 mM NaCl treatments, it showed highly significant and positive correlations with SL (0.94), SFW (0.89), SDW (0.75), and WCP (0.56) and negative with RSR (−0.63). Correlations between shoot- and root-related traits were large under salinity conditions compared to control. The seed viability test was carried out before the start of the experiment, therefore, the correlation coefficients between germination and seedling growth-related traits and viability percentage (SVP) were at control treatment. SVP was significantly correlated with GP, GI, and SVI.

### 2.4. Salinity Tolerance Indices

Five salinity tolerance indices related to GP, SL, SFW, SDW, and WCP were subjected to the analysis of variance between control and each of 75 mM NaCl and 150 mM NaCl treatments (Table 3). Highly significant differences between genotypes were obtained for all salinity tolerance indices in control against both the 75 and 150 mM NaCl treatments. In addition, the highest heritability estimate was found for SLTI (85.8 and 88.1%) for both comparison, respectively. Additionally, the mean values of salinity tolerance indices ranged between 0.72 (SLTI) and 0.98 (WCTI) when control was compared to 75 mM NaCl treatment, while it ranged between 0.52 (SLTI) and 1.31 (SDWTI). The lines S42IL-126, S42IL-132, and S42IL-136 recorded the highest STI for GP in both comparisons as control against 75 mM NaCl and 150 mM NaCl compared to Scarlett (Appendix A). Similarly, lines S42IL-109, S42IL-114, and S42IL-176 were the best lines that showed remarkable STI for seedling length (SL) in both comparisons compared to Scarlett. For SFW, S42IL-109 gave the highest value of SFWTI when control was compared to 75 and 150 mM NaCl treatments, which recorded 1.15 and 1.06, respectively. Meanwhile, lines S42IL-133, S42IL-137, S42IL-142, S42IL-138, and S42IL-176 were amongst lines that revealed higher SDWTI more than unity as compared to Scarlett in both comparisons. For WCP, lines S42IL-148, S42IL-109, S42IL-103, and S42IL-132 were amongst lines that showed superiority in WCTI as compared to Scarlett in both comparisons.

### 2.5. QTL Identification for Germination and Seedling Related Traits

Altogether, 90 significant QTL effects were identified for ten germination- and seedling-related traits distributed over the entire barley genome (Appendix A and Figure 3). Additionally, five QTL were detected for seed viability (Appendix A). Among these, 49 QTL showed introgression line by treatment interaction (*p* < 0.01), ten QTL showed line main effect (*p* < 0.05), and 31 showed both effects. In the following lines, we will present the most important results of QTL detection (Table 4).

#### 2.5.1. Seed Variability and Germination Percentages

Five QTL were detected for SSVP located on chromosomes 1H, 2H, 3H, 5H, and 7H. All these QTLs showed unfavorable relative performance ranging between −16.4 and −20% as compared to Scarlett. Twelve putative QTL were detected for GP distributed on all chromosomes except 1H. Five QTL revealed desirable increase in GP under salinity treatments with relative performance ranged between 22.9 (S42IL-132, 6H under 150 mM NaCl) and 33.3% (S42IL-116, 4H under 75 mM NaCl).

#### 2.5.2. Shoot and Root Lengths

Twenty-two QTL were found for ShL and covered the whole entire barley genome. Based on the relative performance, the QTL S42IL-109 showed strong and desirable increase in ShL by 53.3 and 35.5% as compared to Scarlett under 75 and 150 mM NaCl treatments, respectively. This QTL spans from 33.9–62.7 cM on chromosome 2H. Another QTL (S42IL-110) on 2H spans from 89.5–97.8 cM, exhibited strong and favorable effect for ShL by 50.8% under 75 mM NaCl treatment. Likewise, the QTL analysis revealed seven QTL for RL distributed across chromosomes 2H, 3H, 4H, 5H, and 7H. The ILs, S42IL-109 (2H, 33.9–62.7 cM), S42IL-116 (4H, 1.1–40 cM), S42IL-126 (5H, 76.2–120.3 cM), and S42IL-176 (5H, 81.3–140.1 cM) exhibited desirable positive RP% for RL under one or both salinity treatments and ranged between 30.6 and 40.9%.

#### 2.5.3. Seedling Length and Seedling Vigor Index

Thirteen QTL were detected for SL and distributed on all chromosomes except 1H. Interestingly, four QTL at ILs, S42IL-109 (2H, 33.9–62.7 cM), S42IL-110 (2H, 89.5–97.8 cM), S42IL-115 (3H, 120.7–155 cM), and S42IL-116 (4H, 1.1–40 cM) exhibited desirable significant increase by RP ranging between 28.4 and 42.1% in SL under 75 mM NaCl conditions as compared to Scarlett. Also, two QTLs on chromosomes 2H and 6H were detected for SL, since S42IL-109 and S42IL-132 showed remarkable increase in SL by RP values 30.9 and 31.4% under 150 mM NaCl conditions. Notably, S42IL-109 showed significant increase in SL under both salinity treatments. For SVI, fourteen QTL were identified and covered the entire barley genome. Eight QTL out of them revealed desirable increase in SVI under salinity conditions by RP values ranging between 35 (S42IL-148, 6H) and 73.2% (S42IL-116, 4H) under 75 mM NaCl treatments, and between 45 (S42IL-109, 2H) and 61.5% (S42IL-132, 6H) under 150 mM NaCl treatments.

#### 2.5.4. Seedling Fresh and Dry Weights

The QTL analysis revealed four QTL for SFW and were distributed on chromosomes 2H and 6H. The strongest QTL was found in S42IL-132 located on chromosome 6H within the interval 94.9 and 108.3 cM, which resulted in 40.6% increase in SFW under 150 mM NaCl conditions as compared to Scarlett. Only one QTL was detected for SDW of the S42IL-107 (2H, 12.5–41.2 cM), which resulted in −56% decrease in SDW under 75 mM NaCl treatments.

#### 2.5.5. Root/Shoot Ratio and Water Content Percentage

Fifteen QTL were identified for RSR and mapped on chromosomes 1H, 2H, 4H, 5H, 6H, and 7H. All detected QTL showed remarkable increase in RSR under both salinity treatments. The strongest QTL effect under 75 mM NaCl conditions was found in S42IL-125 (5H, 51.5–81.3 cM), which resulted in a 121.1% increase in RSR. While under 150 mM NaCl conditions, the strongest QTL effect was obtained for S42IL-127 (5H, 138.5–162.5 cM), which accounted for a 304.6% increase in RSR as compared to Scarlett. Two QTL were identified for WCP and distributed on chromosome 2H. The QTL in the S42IL-109 (2H, 33.9–62.7 cM) showed remarkable increase in WCP under 75 mM NaCl conditions by RP value of 7.3% as compared to Scarlett. WCP

### 2.6. QTL Detection for Salinity Tolerance Indices

#### 2.6.1. Germinability Tolerance Index (GTI)

Eleven QTL were identified for the two levels of GTI and distributed on chromosomes 4H, 5H, 6H, and 7H (Table 5 and Figure 3). All QTLs showed a remarkable increase in GTI by RP% ranged between 28.4 and 39.4% as eight QTLs under 75 mM NaCl, three under 150 mM NaCl, and one QTL under both treatments. The strongest QTL was found in S42 IL-148 (6H, 0.3–11.3 cM), which resulted in a 39.4% increase in GTI as compared to Scarlett under 75 mM NaCl, while under 150 mM NaCl, the strongest QTL effect was obtained for S42 IL-135 (7H, 67.8–118.5 cM), which gave a 32% increase in GTI as compared to Scarlett.

#### 2.6.2. Seedling Length Tolerance Index (SLTI)

Ten QTLs were detected for SLTI and located on chromosomes 2H, 3H, 5H, 6H, and 7H (Table 5 and Figure 3). The introgression lines (QTLs); S42 IL-109 (2H, 33.9–62.7 cM) and S42 IL-176 (5H, 81.3–140.1 cM) exhibited positive RP% by values of (93.9 and 78.5%) and (53.4 and 54.7%) under 75 and 150 mM NaCl treatments, respectively.

#### 2.6.3. Seedling Fresh (SFWTI) and Dry Weights Tolerance Indices (SDWTI)

Two QTL were found for SFWTI and mapped on chromosome 2H (Table 5 and Figure 3). The strongest QTL was found in S42 IL-109 (2H, 33.9–62.7 cM) and showed IL main effect. Interestingly, the exotic alleles at this locus increased SFWTI by RP% values of 73.7 and 71.7% under both treatments, respectively. In addition, the second QTL was found for the IL S42 IL-107 (2H, 12.5–41.2 cM), which showed increase in SFWTI under 150 mM NaCl conditions as compared to Scarlett. Two QTL for SDWTI as line main effect and mapped on chromosome 7H. Both QTL *QSdwti.S42IL.7H.a* and *QSdwti.S42IL.7H.b* displayed desirable relative performance with values of 64.2 and 59.1%, respectively.

#### 2.6.4. Water Content Percentage Tolerance Index (WCPTI)

Ten QTLs were detected for WCPTI and distributed on chromosomes 2H, 3H, 4H, 5H, and 6H (Table 5 and Figure 3). All these QTLs exhibited ILs by treatment interaction, as nine QTLs under 75 mM NaCl conditions and one QTL under 150 mM NaCl conditions. The strongest QTL effect was found in S42 IL-109 (2H, 33.9–62.7 cM), which resulted in a 10.9% increase in WCPTI, followed by S42 IL-147 (5H, 73.3–81.3 cM), which gave a 10% increase in WCPTI under 75 mM NaCl conditions. Additionally, the only QTL that was detected under 150 mM NaCl conditions was found in S42 IL-153 (2H, 60.7–68.6 cM), which resulted in a decrease in WCPTI by RP% value of −12.4%.

## 3. Discussion

Germination and seedling growth are considered critical stages in the plant life cycle, especially under unfavorable environmental conditions [27,28,29]. In many production areas worldwide, particularly arid and semi-arid environments, barley is affected by salinity stress, which reduces both grain yield and quality [30,31]. However, salinity tolerance in crops is a complex quantitative trait both genetically and physiologically, and any induced change by salinity is modulated by changes in gene expression [32,33]. So, uncovering and understanding the genetic makeup is of paramount importance to improving salt tolerance. Identifications of desirable candidate genes or QTL associated with salinity tolerance is becoming increasingly important for plant breeders and consequently for farmers. Such stress tolerance improvement can be achieved through genetic variation that is present in crop wild relatives, which can be a major source in providing the required genes since they show a huge allelic variation, often much wider than within a crop gene pool [21]. The current study aimed at detecting QTL effects of the exotic alleles in a set of 50 ILs carrying ISR42-8 introgressions in the Scarlett background to salinity tolerance at germination and seedling phases. Since the wild barley accession ISR42-8 contains detrimental alleles for both agronomic and adaptive traits, the S42 population and its ILs displayed transgression segregations for these traits under control and adverse conditions, referring to presence of the valuable alleles in the wild gene pool [34,35,36,37,38,39,40].

In the current study, salinity stress significantly affected barley seed germination process and seedling growth-related traits, causing a clear reduction in all traits, except MGT, RSR, and SDW which were increased in most of the ILs, and this is because of the adverse impacts of salinity in barley growth [41]. The high salinity treatment applied in this study (150 mM NaCl) caused a 11.3% reduction in GP. Angessa et al. [42] reported a 18% reduction in GP of barley seeds germinated on 150 mM NaCl. The observed reduction in germination, may be attributed to a salty environment that increases the osmotic pressure that creates a condition similar to drought, therefore the seeds were exposed to salt- induced physiological drought stress. This process impairs the ability of seeds to absorb water from the germination medium, hence prolonging or even inhibiting seed imbibition and subsequently germination and plant growth [43,44]. We demonstrated a wide range of salinity tolerance in the 50 S42ILs at germination stage. The S42IL-148 is a salinity sensitive genotype, which had the lowest GP of 57.8% with a reduction of 25.7% compared to Scarlett in 150 mM NaCl treatment. In contrast, some S42ILs such as S42IL-122, S42IL-118, and S42IL-109 were less affected by severe salt treatment and had high germinability and took less time to germinate under salinity conditions compared to Scarlett, indicating their osmotolerance during germination stage. This salt-tolerance may be due to that the seeds increased their osmotic potential through sodium uptake from the germination medium that led to absorbing more water under salinity stress [45].

In addition, remarkable differences for ShL, RL, SL, RSR, SFW, SDW, and WCP were detected amongst S42Ils set under control and salinity conditions, referring to the segregation of selected traits across 50 ILs. We found that ILs S42IL-109, S42 IL-120, S42 IL-125, S42 IL-126, S42 IL-127, S42IL-132, S42 IL-148, and S42IL-176 showed significant superiority in seedling growth-related traits under both salinity treatments compared to Scarlett, indicating osmotolerance at seedling phase. This result was confirmed by salinity tolerance indices. Furthermore, increasing RSR under salinity conditions may be attributed to the rapid reduction of shoot dry matter production for consequent shortening shoot length and elongating roots for searching water [46]. These salt-tolerant ILs could be suggested to be used in breeding programs to develop new elite barley cultivars. Consistent with our findings, different parameters related to seed germination and seedling growth were significantly affected by salt stress [15,16,18,47,48,49].

The significant differences, which were observed among treatments and for G by T interactions, referred to the obvious effect of salt stress on the tested genotypes, which responded differently to the levels of salt concentration. Such differences have been observed for the same group of S42ILs under different environmental conditions such as drought and nutrient deficiency [21,35,37,38,39,47]. Moreover, this noticeable variation among the genotypes in the current study was reflected through the high estimates of broad-sense heritability under salinity conditions in the current study. The resultant heritability estimates of investigated traits were high and stable across salinity treatments, which is important for selection in barley breeding programs. This finding indicates that salinity tolerance along with germination- and seedling-related biological processes are genetically controlled, which could be explained by QTL analysis. Confirming our finding, moderate to high estimates of heritability for salt-tolerance-related traits at germination and seedling stages were obtained in other reports [10,15,16,48,49,50,51] and in the same S42Ils library under phosphorus deficiency [21] and water stress [35].

### 3.1. S42ILs by Trait Associations

By association analysis, altogether 104 significant line by trait associations, summarized to 90 putative QTL due to the overlapping or flanking of several introgressions in the S42IL set. At forty QTL (44.4%), the exotic introgression alleles were associated with an improved trait performance under salinity conditions, indicating that selection for salt tolerance alleles could be achieved. Consistent with this result, Schmalenbach and Pillen [48] detected 40 QTL for eight malting quality traits using S42ILs. Also, Arifuzzaman et al. [49] identified nine QTL effects of the exotic alleles, which were verified using a library of 53 S42ILs for shoot and root traits. Naz et al. [35] detected 15 chromosomal regions where the exotic QTL alleles of the S42ILs showed improvement for root and related shoot traits under drought conditions. In contrast, Honsdorf et al. [39] found that *Hsp* allele had a trait reducing effect of 26 QTL out of 40 under drought stress.

### 3.2. Germination Percentage Linked QTL

The exotic alleles in the S42IL-116 (4H), S42IL-132 (6H), S42IL-122 (6H), and S42IL-136 (7H) genotypes had a positive effect on the germination performance under salinity conditions as line main and line by salinity interactions effects. These genotypes exceeded Scarlett in the germination percentage under salinity conditions by 33%. To our knowledge, the current study is the first report that has evaluated S42ILs for germination under salinity conditions, hence, we could not detect corresponding QTL for the trait performance in the S42IL library or even S42 population. Mano and Takeda [17] detected significant QTL for salt tolerance at germination on chromosomes 4H, 5H, 6H, and 7H using two doubled haploid barley populations derived from the crosses, Steptoe/Morex and Harrington/TR306. In addition, unfavorable effects of the *Hsp* allele in the S42IL-107 on chromosome 2H (12.5–41.2 cM) on GP under both control and salinity conditions were observed. Angessa et al. [42] identified two QTL on 2HL (156–160.2 cM) linked with GP in 150 and 300 mM NaCL in barley.

### 3.3. Seedling Growth-Related Traits Linked QTL

In most cases, the exotic alleles of the detected QTL had unfavorable effects on seedling growth traits under salinity conditions. Several ILs exhibited pleiotropic effects, but few showed a desirable performance. Interestingly, a genomic region within the interval 33.9–62.7 cM on chromosome 2H in the S42IL-109 was associated with ShL, RL, SL, SVI, and WCP. The exotic alleles at this QTL showed an exceptional desirable performance under salinity conditions compared to Scarlett. Naz et al. [35] found that the genotype S42IL-109 was significantly linked to root dry weight and tillers number under drought stress. Soleimani et al. [21] found that the *Hsp* allele in line S42IL-109 increased shoot:root length ratio by 30.13% under phosphorus deficiency. Additionally, Honsdorf et al. [39] stated that this genotype was associated with reducing plant height along with biological and grain yields under drought conditions. In another study, Arifuzzaman et al. [49] reported that this genotype revealed significant decrease in plant height as compared to Scarlett under contrasting water regimes conditions due to the contribution of the *Hsp* alleles. Sayed et al. [40] detected a QTL *QRL.S42-2H* (25-7-30.2 cM) for RL at seedling stage and the *Hsp* alleles increased RL by value of 26.4% compared to Scarlett. Based on the above results, this genomic region may have genetic factors that influence seedling growth-related traits in the S42IL-109.

Similarly, the genotype S42IL-176 bore a QTL contributing in increasing RL by 35.2 and 30.6% in the 75 and 150 mM NaCl treatment induced, respectively. This QTL (*QRl.S42IL.5H.b*) locates between 81.3–140.1 cM on chromosome 5H and co-locates to *QYld.S42.5H, QTgw.S42.5H,* and *QSpad.S42.5H* as reported by Zahn et al. [38] and was very close to the *QRL.S42-5H* (150–162 cM) as reported by Sayed et al. [40]. Naz et al. [26,35] found that S42IL-176 was associated with root-related traits including RL under drought stress. Also, Soleimani et al. [21] stated that this genotype associated significantly with RL under phosphorus deficiency. Interestingly, the *VrnH1* gene, which spans between 94.95–126.76 cM, was significantly associated with RL, and the *Hsp* alleles displayed about a 9% increase in RL [49]. Similarly, Sayed et al. [36] detected a QTL on 5H (95 cM), where the exotic allele enhanced proline content by 54%.

A QTL on 6H (94.9–108.3 cM) showed pleiotropic effects and was linked to increasing each of SL, SVI, SFW, and GP in the IL S42IL-132 under 150 mM NaCl treatment. This IL seems to be a promising genomic region carrying a valuable novel allele for salinity tolerance. Also, the ILs S42 IL-141 and S42 IL-143 on 1H showed favorable line main effect of RSR, these two lines harboring a *Pyrroline-5-carboxylate synthase1- P5cs1* allele derived from the wild barley accession ISR42-8 which showed high proline accumulation [50], and displayed less severe wilting under drought stress [25].

By contrast, the detected QTL on chromosome 2H between 12.5–41.2 cM in the S42IL-107 showed unfavorable performance of traits GP, SL, SVI, SFW, and SDW under control and salinity conditions, which displayed high reduction percentages. This genotype reduced each number of grains per spike, biomass, grains weight per spike, plant height, and heading time under drought stress [39]. Also, this genotype was associated with reducing grain yield, grains per spike, and increasing chlorophyll content under low nitrogen input [38]. Interestingly, in the S42IL-107, the *Hsp* allele had negative effects on biomass, which led to 43% loss in biomass in the adult plants [39], while at the juvenile development stage, this genotype showed increase in biomass production under severe drought stress [37]. Xu et al. [51] identified a significant QTL for salinity tolerance on chromosome 2H at a position of 14.7 cM. Notably, this genotype contains the flowering gene *Ppd-H1* that causes early flowering in barley [39]; however, it seems to have additional effects on seedling-related traits. Similarly, the exotic alleles in the S42IL-153 on 2H (60.7–68.6 cM) had undesirable effects on ShL, RL. SL, SVI, SFW, and WCP performance under salinity conditions. This location was associated with the reduction of nitrogen and chlorophyll contents in barley ILs [24].

The present findings also brought out conclusive evidence for the genetic control of salinity tolerance indices of the traits GP, SL, SFW, SDW, and WCP among the 50 ILs, which was reflected by the high heritability estimates and the wide range of STIs that was observed among the genotypes. Additionally, this suggestion was confirmed by the least square means between ILs and Scarlett through Dunnett test. Across and under salinity treatments, the QTL in the ILs S42IL-121, S42IL-126, S42IL-127, S42IL-132, and S42IL-135 revealed the highest favorable RP% compared to Scarlett. According to STI, these genotypes are salinity tolerant at germination stage. In addition, the ILs S42IL-109, S42IL-132, and S42IL-176 exhibited the highest salinity tolerance for SL in 150 mM NaCl treatment. Likewise, the S42IL-109 and S42IL-148 revealed desirable salinity tolerance index for WCP. Our findings agreed with other studies, Thabet et al. [52] and Mwando et al. [10] detected genes associated with salt-tolerance index at germination and seedling stage in barley.

### 3.4. QTL Validation for Salt Tolerance

Recently, Sayed et al. [53] evaluated the S42 population for grain weight, and its attributes under severe salinity conditions in Egypt, and detected 49 QTL for grain weight, its attributes, and their salinity tolerance indices. We validated seven exotic QTL alleles in the 50 ILs set used in the current study. The ILs S42IL-127 (5H, 138.5–162.5 cM) and S42IL-139 (7H, 129.5–141.1 cM) were associated with reducing ShL and corresponding to the QTL *qPH.5H.b* (AF043094A, 5H, 156 cM) and *qPH.7H* (bPb-1793, 7H, 137.2 cM) where the *Hsp* alleles reduce plant height (PH), which were detected by Sayed et al. [53]. In addition, IL S42IL-125 (5H, 51.-81.3 cM), which increases RSR, corresponds to the QTL *qPH.5H.a* (Bmag357, 5H, 68 cM) where the exotic allele increased PH [53]. Notably, this QTL harbored an *Hsp* allele correlated with QTL for RL, RDW, and RSR [49]. These results showed the importance of these QTL related to plant height. The QTL related to the ILs S42IL-117, 118, and 121 span 17.8–81 cM on 4H, showed reduction in seedling growth-related traits and corresponded to *HVPAZXG* (44 cM) and *bPb-6640* (60.5 cM) on 4H, in which the *Hsp* allele delays heading time under salinity conditions. The majority of detected novel QTL linked to SFW and SDW and their STI were mapped on 2H and 6H and did not show any correspondence to the QTL detected for grain and biological weights under salinity conditions. Only one QTL on 7H within the interval 86–127.5 cM in the S42IL-137 was associated with increasing SDWTI and was corresponding to the DArT marker *bPb-5260* for SFWTI and SDWTI were mapped on 2H. Naz et al. [49] validated nine putative exotic QTL alleles for drought tolerance in a set of ILs including S42IL-109, S42IL-137, and S42IL-148, which were associated with PH. However, many studies have been carried out at different developmental stages in barley to study the effects of salt on plant growth at genetic, physiological, and morphological levels [45,52,54,55,56,57]. The results suggest that detected QTL dominating salt tolerance at early growth stages differ from those regulating the same response in older plants, confirming that different abiotic stresses are stage-specific [58,59]. The validated QTL linked to salinity tolerance in the current study, corresponding and integrated to previously published QTL under different abiotic stresses, suggest the stability of these QTL effects across variable environmental conditions. Interestingly, the results confirmed that the QTL alleles that control salinity tolerance at germination differ from those in the seedling stage. Similar findings were obtained by Mano and Takeda [17].

Use of wild relatives to improve salinity tolerance is common in field crops. For instance, the presence of the Na^+^ exclusion gene *TmHKT1*; 5-A (*Nax2*), encoding a Na^+^-selective transporter, increased grain yield on saline soils in durum wheat by up to 25% [60]. Qi et al. [61] identified a novel ion transporter gene, GmCHX1, and related its sequence alterations to salt tolerance in wild soybean. On the basis of our QTL study, several QTL (ILs) where the exotic *Hsp* allele had a desirable effect on trait performance under salinity conditions were identified. Particularly, S42IL-109 (2H), S42IL-116 (4H), S42IL-132 (6H), S42IL-133 (7H), S42IL-148 (6H), and S42IL-176 (5H) are promising candidates to improve salinity tolerance at germination and seedling stages, and these ILs might be very interesting for further breeding. Additionally, the results confirm the potentiality of the wild progenitor genes to improve barley performance under salinity conditions, which can be cloned for the most promising QTL detected in our study, as a genomic tool available for barley breeding and genetics. Honsdorf et al. [37,39], Arifuzzaman et al. [49], and Naz et al. [26] recommended using the most promising ILs such as S42Il-121 and S42IL-176 in barley breeding programs and to transfer them in the cultivated barley. von Korff et al. [62] noted the importance of the ILs in the assessment and utilization of exotic barley with the purpose to promote breeding progress and enable studies of gene function, in addition to locating new DNA markers on the genome and for fine mapping of genes.

## 4. Materials and Methods

### 4.1. Plant Material and Genotyping

Fifty wild barley ILs of the S42IL library (83 lines) and the recipient parent Scarlett were selected for the experiment. The S42IL library was developed from the advanced backcross population S42 and described in detail by Schmalenbach [47,63]. In brief, the S42ILs were originated from the cross between the German spring cultivar Scarlett (*Hordeum vulgare* ssp. *vulgare*) and the wild barley accession ISR42-8 (*H. vulgare* ssp. *spontaneum*, Koch), followed by three rounds of backcrossing to Scarlett as a recurrent parent and several rounds of self-pollination, combined with marker-assisted selection to produce a BC_3_S_4_ population (83 lines). Each line includes a single marker-defined chromosomal segment of the wild barley accession ISR42-8, meanwhile the remaining part of the genome is derived from the elite barley cultivar Scarlett. The introgression lines were genotyped with Illumina 9K SNP chip and genetic map of the S42ILs library was created by Comadran et al. [64], where more details can be found. The genetic characterization of position and extent of *Hsp* introgressions of the complete set based on the Infinium 9k iSelect assay has been published by Honsdorf et al. [39].

### 4.2. Seed Viability

The seed viability test was carried out according to the International Seed Testing Association [65] as follow: A sample of 10 grains from each genotype replicated three times was immersed in H2O for 18 h, at a 20 °C temperature, then the seeds were stained with 2, 3, 5 triphenyl tetrazolium chloride, by immersing seeds in 1% solution for 3 h, in the absence of light and at a temperature of 30 °C. Afterward, a longitudinal cross-section of the embryo was made to split the seed into two halves, the number of stained seeds was counted. Seed viability percentage (SVP; %) was calculated as follow:$$\mathrm{SVP}\;(\%)=\frac{\mathrm{Number}\;\mathrm{of}\;\mathrm{stained}\;\mathrm{seeds}}{\mathrm{Total}\;\mathrm{number}\;\mathrm{of}\;\mathrm{seeds}}\times 100$$

### 4.3. Salinity Stress Treatments

The experiment was conducted at the Leibniz Institute of Plant Genetics and Crop Plant Research (IPK), Gatersleben, Germany in a completely randomized design. A set of 45 seeds of each genotype was surface sterilized with 70% ethanol solution for one minute, and rinsed with sterile distilled water several times, then briefly blotted. The seeds were placed on two layers of filter papers (C160; Ahlstrom-Munksjö, GmbH, Dettingen, Germany) in crystal clear rectangular boxes (V3-92; Licefa GmbH & Co. KG, Bad Salzuflen, Germany). The salinity treatments, with three replicates were conducted by watering the seeds with two different concentrations, 75 and 150 mM NaCl (Sodium chloride CELLPURE^®^ ≥99.5%, for cell culture and biochemistry, Carl Roth, GmbH, Karlsruhe, Germany), whereas deionized water was applied as a control treatment, and the seeds were placed in a versatile environmental test chamber (Model No. MLR-352-PE, Panasonic, Osaka, Japan) for ten days, maintained at 20 ± 2 °C with 50 ± 5% humidity at 12 h light (200 μmol m^−2^ s^−1^) and 12 h dark periods per day. The seeds were considered germinated when the radicle reached at least 2 mm in length and the number of the germinated seeds was counted daily after 24 h of incubation until the end of the experiment. The experiment was repeated twice for the salinity treatments using identical conditions, thus, a total of 135 seeds of each of the S42ILs and Scarlett were evaluated for salinity tolerance at germination and seedling stage.

### 4.4. Evaluation of Germination Parameters

Seeds were counted daily until the 10th day in order to calculate the following parameters.

Germination percentage (GP; %):
$$\mathrm{GP}\;(\%)=\frac{\mathrm{Number}\;\mathrm{of}\;\mathrm{germinated}\;\mathrm{seeds}}{\mathrm{Total}\;\mathrm{number}\;\mathrm{of}\;\mathrm{sowed}\;\mathrm{seeds}}\times 100$$Germination index (GI) was calculated according to Ranal [66] as follows:
GI = (10 × N1) + (9 × N2) +…+(1 × N10); 
where, N1, N2…N10, is the number of seeds germinated on the first, second and subsequent days until 10th day and the multipliers (i.e., 10, 9…etc.) are weights given to the days of the germination.Mean germination time (MGT) was calculated according to Mudaris [67] as follows:
MGT = Σ(Ti × Ni)/ΣNi
where, Ni is number of the new germinated seeds at time Ti.

### 4.5. Evaluation of Seedling Growth Parameters

Shoot length (ShL) and root length (RL) in cm were measured manually at the tenth day of germination using a scaled ruler for five seedlings from each replicate at the end of the experiment, and both lengths were summed to obtain the total seedling length (SL; cm). Root–shoot ratio (RSR) was calculated as the ratio of the RL to the ShL. The seedlings fresh weight (SFW) was recorded (g) using an ultra-micro lab balance (Sartorius AC 1215, Germany), then seedlings were dried at 80 °C for 72 h to obtain the seedling dry weight (SDW). Seed vigor index (SVI) was calculated by multiplying germination percentage and seedling length [68]. Water content percentage (WCP; %) was calculated based on the following formula:$$\mathrm{WCP}\;(\%)=\frac{\left(\mathrm{SFW}-\mathrm{SDW}\right)}{\mathrm{SFW}}\times 100$$

### 4.6. Stress Tolerance Index (STI)

In order to evaluate the growth performance and the variation among genotypes in their tolerance to salinity, stress tolerance index (STI) was employed for the following parameters; germination percentage (GPTI, as germination percentage tolerance index), seedling length (SLTI, as seedling length tolerance index), seedling fresh weight (SFWTI, as seedling fresh weight tolerance index), seedling dry weight (SDWTI, as seedling dry weight tolerance index), and water content % (WCPTI, as water content percentage tolerance index). The salt tolerance indices (STIs) for these traits were calculated according to the below formula of Fernandez [69]:$$\mathrm{STI}\;=\frac{\mathrm{Trait}\;\mathrm{value}\;\mathrm{under}\;\mathrm{salt}\;\mathrm{treatment}}{\mathrm{Trait}\;\mathrm{value}\;\mathrm{under}\;\mathrm{control}}$$

### 4.7. Statistical Analyses

The separate and combined analysis of variance (ANOVA) of a completely randomized experiment were performed using SAS software v. 9.2 with PROC GLM procedure [70], to test the effect of each treatment and the interaction between the IL and the treatments. Broad-sense heritability (*H_b_*) estimates were calculated under control and salinity conditions following Padi [71].
$$H_{b}=\frac{\sigma_{g}^{2}}{\sigma_{p}^{2}},\;\;\;\;\;\;\;\;\;\;\;\;\;\;\;\sigma_{p}^{2}=(\sigma_{g}^{2})+\left(\frac{\sigma_{e}^{2}}{r}\right)$$
where, $\sigma_{g}^{2}$ is genotypic variance, $\sigma_{p}^{2}$ is phenotypic variance, $\sigma_{e}^{2}$ is pooled error variance, and r is number of replicates.

Additionally, least squares means (Lsmeans) were calculated for each genotype using PROC GLM method of SAS software. The phenotypic Pearson Correlation matrix analysis among the traits in control and 150 mM NaCl treatments was calculated by R-studio.

### 4.8. QTL Detection

To detect putative QTL, the Dunnett test was performed using SAS software [70] to test the most significant differences between the individual introgression lines and Scarlett as a control [36,40]. QTL were classified according to the significance level, if the phenotypic Ls-mean of the introgression line (IL) was significantly different (*p* < 0.01) to Ls-mean of Scarlett in one or two treatments, it was considered as line × treatment interaction, while if the IL was significantly different (*p* < 0.05) to Scarlett in the three treatments or overall treatments, it was classified as a line main effect. Following Honsdorf et al. [39] and Naz et al. [35], when the S42IL shows a significant difference to Scarlett, it is assumed that a QTL is present within the *Hsp* introgression of that line, and if two independent S42ILs contained overlapping *Hsp* introgression and both lines exhibited a significant trait effect with the same sign, it is assumed they contained the same QTL. The QTL analysis was done for all investigated traits except germination index (GI) and mean germination time (MGT).

## 5. Conclusions and Outlook

At forty (44.4%) QTL out of 90, the *Hsp* introgression alleles are involved in improving salinity tolerance at germination and seedling growth stage. Among them, seven exotic QTL alleles were successfully validated in wild barley ILs. However, it is desirable to evaluate the S42ILs library under saline conditions in the field for further validations in the adult plants. The present study provides new S42ILs like S42IL-109 (2H), S42IL-116 (4H), S42IL-132 (6H), S42IL-133 (7H), S42IL-148 (6H), and S42IL-176 (5H) as valuable genetic resources, and consequently novel QTL for further enhancement of salt tolerance at germination and seedling development stages in barley breeding. However, further studies on functional characterization by utilizing the agronomic, genetic, physiological, and biochemical indicators are needed that can facilitate and prove the identification of the salinity tolerance genes at early growth stages.

## Figures and Tables

**Figure 1 plants-10-02246-f001:**
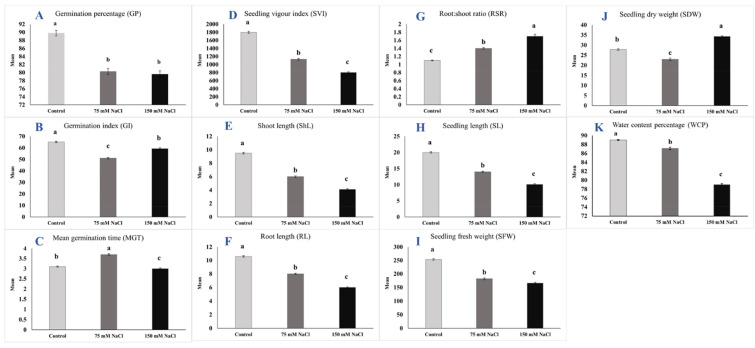
Means with standard errors of all studied traits under control, 75 mM NaCl, and 150 mM NaCl conditions (different letters mean significant differences). Note: (**A**) GP is germination percentage (%), (**B**) GI is germination index, (**C**) MGT is mean germination time (day), (**D**) SVI is seedling vigor index, (**E**) ShL is shoot length (cm), (**F**) RL is root length (cm), (**G**) RSR is root:shoot ratio, (**H**) SL is seedling length (cm), (**I**) SFW is seedling fresh weight (mg), (**J**) SDW is seedling dry weight (mg), and (**K**) WCP is water content percentage (%).

**Figure 2 plants-10-02246-f002:**
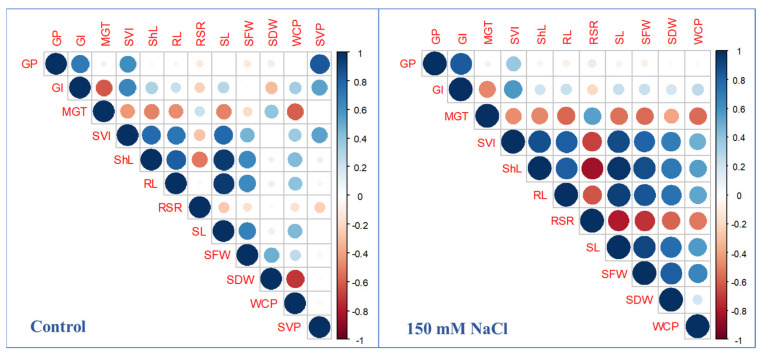
Pearson correlation coefficients among studied traits under control (**left**) and 150 mM NaCl conditions (**right**). Where, GP is germination percentage (%), GI is germination index, MGT is mean germination time (day), SVI is seedling vigor index, ShL is shoot length (cm), RL is root length (cm), RSR is root: shoot ratio, SL is seedling length (cm), SFW is seedling fresh weight (mg), SDW is seedling dry weight (mg), WCP is water content percentage (%), and SVP is seed viability percentage (%).

**Figure 3 plants-10-02246-f003:**
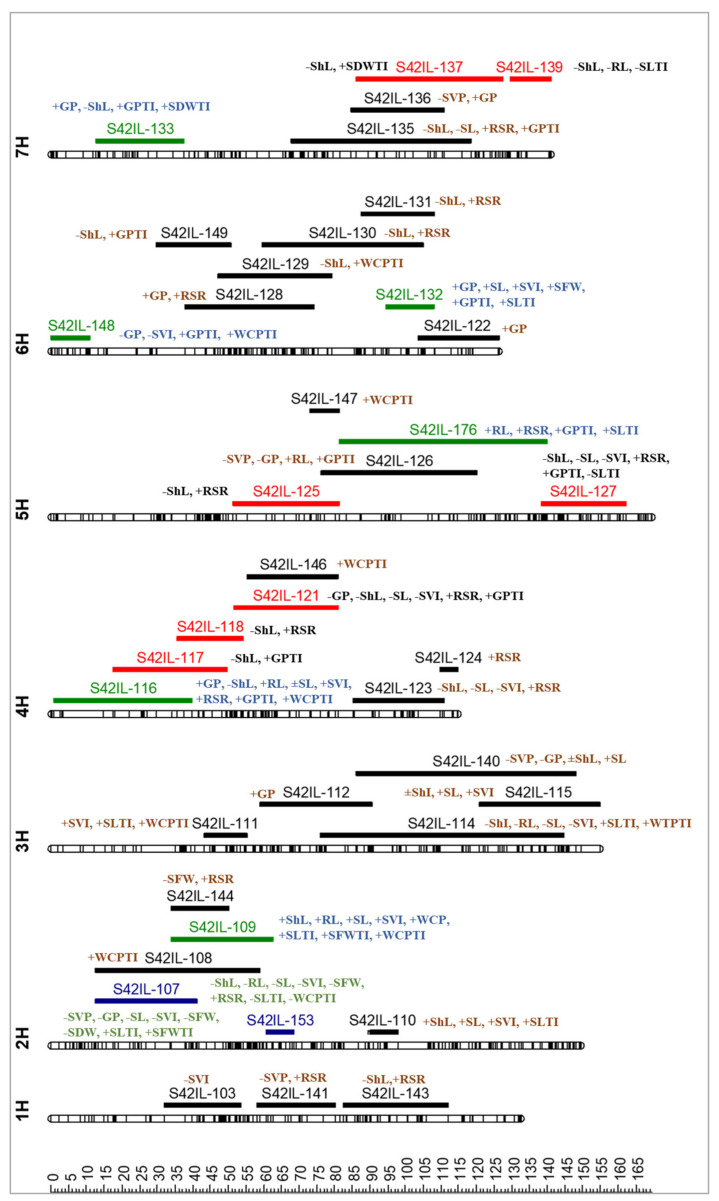
Localization of the most significant QTL in the S42ILs associated with germination- and seedling growth-related traits under control and salinity treatments. QTL are placed around the S42ILs, indicated by trait abbreviations. The sign indicates an increasing (+) or decreasing (−) *Hsp* effect. Green ILs indicate the most favorable effects of *Hsp* alleles, blue ILs indicate the most unfavorable effects of *Hsp* alleles, and red ILs indicate verified and validated QTL with previously published QTL in adult barley plants.

**Table 1 plants-10-02246-t001:** Separate and combined ANOVA, coefficient of variation (C.V.%), coefficient of determination (R^2^), and broad-sense heritability (H_b_) for the investigated traits in the current study.

Trait	Control				75 mM NaCL				150 mM NaCL				Combined ANOVA		
	MS	C.V.%	R^2^	H_b_	MS	C.V.%	R^2^	H_b_	MS	C.V.%	R^2^	H_b_	G	T	G × T
GP	172.49 **	5.5	0.78	85.98	196.75 **	7.2	0.74	82.98	244.55 **	8.1	0.74	83.02	346.6 **	148.8 **	133.6 **
GI	200.3 **	5.0	0.90	94.71	145.96 **	10.3	0.72	81.14	174.22 **	10.4	0.69	78.01	277.7 **	7583 **	121.4
MGT	0.47 **	5.5	0.88	93.62	0.54 **	9.9	0.66	75.02	0.34 **	9.2	0.68	77.36	0.44 **	20.5 **	0.46 **
SVI	190,565 **	10.9	0.71	80.00	166,431 **	11.9	0.82	89.05	134,821 **	13.4	0.85	91.38	163,643 **	39,506,781 **	164,087 **
ShL	4.44 **	11.2	0.66	74.76	6.44 **	13.1	0.83	90.24	6.79 **	16.8	0.87	92.98	5.1 **	1120.1 **	6.3 **
RL	4.15 **	10.5	0.62	70.40	2.66 **	9.9	0.67	76.18	2.98 **	10.5	0.79	86.85	3.02 **	815.1 **	3.4 **
RSR	0.02 **	11.0	0.42	30.92	0.24 **	13.1	0.78	86.41	1.03 **	21.5	0.80	87.63	0.47 **	10.6 **	0.41 **
SL	15.65 **	9.4	0.68	77.17	15.16 **	10.1	0.79	86.75	17.21 **	11.2	0.87	92.61	13.5 **	3839.5 **	17.3 **
SFW	2166.4 **	12.0	0.53	57.39	3972.9 **	16.9	0.67	76.37	2749.5 **	14.3	0.70	79.39	2875.9 **	328,179 **	3006.5 **
SDW	48.32 **	18.0	0.49	48.59	102.15 **	21.4	0.67	76.27	77.80 **	17.7	0.51	52.65	63.5 **	4981.3 **	82.4 **
WCP	5.66 **	1.4	0.63	71.09	26.47 **	2.4	0.75	83.37	18.00 **	3.2	0.57	63.69	14.5 **	4294.6 **	17.8 **
SVP	123.50 *	6.5	0.66	74.99											

Where, GP is germination percentage (%), GI is germination index, MGT is mean germination time (day), SVI is seedling vigor index, ShL is shoot length (cm), RL is root length (cm), RSR is root:shoot ratio, SL is seedling length (cm), SFW is seedling fresh weight (mg), SDW is seedling dry weight (mg), WCP is water content percentage (%), and SVP is seed viability percentage (%). MS refers to mean square of trait interest, G is genotype, T is treatment, and G × T is genotype by treatment interaction. * and ** significant and highly significant at 0.05 and 0.01 probability levels, respectively.

**Table 2 plants-10-02246-t002:** Summary statistics and reduction percentage (R%) due to salinity effects on the studied traits as compared to control.

Trait	Control		75 mM NaCl			150 mM NaCl			
	Mean ± S.E.	Range	Mean ± S.E.	Range	R% ^a^	Mean ± S.E.	Range	R% ^b^	R% ^c^
GP	89.8 ± 0.69	60–100	80.3 ± 0.75	60–100	−10.6	79.6 ± 0.84	53.3–100	−11.4	−0.9
GI	65.1 ± 0.69	45–85	51.1 ± 0.65	33–68	−21.5	59.3 ± 0.73	37–79	−8.9	16.0
MGT	3.1 ± 0.03	2.3–4.2	3.7 ± 0.04	2.45–5.08	19.4	3.3 ± 0.03	2–4.36	−3.2	−18.9
SVI	1799.1 ± 24.01	1026–2470	1132 ± 20.9	553.3–1800	−37.1	801.6 ± 18.4	271.1–1466.6	−55.4	−29.2
ShL	9.5 ± 0.12	5–13.7	6 ± 0.12	2.2–10.47	−36.8	4.1 ± 0.12	0.6–8	−56.8	−31.7
RL	10.6 ± 0.11	6.5–15.33	8 ± 0.09	4.7–11.1	−24.5	6 ± 0.09	2.83–8.33	−43.4	−25.0
RSR	1.1 ± 0.01	0.8–1.67	1.4 ± 0.02	0.91–3	27.3	1.7 ± 0.05	0.91–5.56	54.5	21.4
SL	20 ± 0.22	12.2–29	14 ± 0.20	8.3–20.33	−30	10.1 ± 0.21	3.83–15.6	−49.5	−27.9
SFW	253 ± 2.95	165.7–351.7	181.8 ± 3.55	58.4–272	−28.2	166.1 ± 2.89	35.3–241.7	−34.4	−8.6
SDW	27.8 ± 0.46	12.8–43.5	23 ± 0.57	7.6–40.5	−17.3	34.3 ± 0.34	11.93–48.97	23.4	49.1
WCP	89 ± 0.13	86.4–94.9	87.1 ± 0.27	80.2–92.7	−2.1	79 ± 0.25	61.68–87.29	−11.2	−9.3
SVP	85 ± 0.63	65–100							

Where, GP is germination percentage (%), GI is germination index, MGT is mean germination time (day), SVI is seedling vigor index, ShL is shoot length (cm), RL is root length (cm), RSR is root: shoot ratio, SL is seedling length (cm), SFW is seedling fresh weight (mg), SDW is seedling dry weight (mg), WCP is water content percentage (%), and SVP is seed viability percentage (%). R% ^a^ and R% ^b^ are reduction percentages for all studied traits occurred in 75 mM and 150 mM NaCl treatments compared to control treatment. R% ^c^ refers to reduction percentage occurred in 150 mM NaCl compared to 75 mM NaCl treatment.

**Table 3 plants-10-02246-t003:** Mean squares (MS), coefficient of determination (R^2^), and broad-sense heritability (H_b_), means ± standard error (S.E.), coefficient of variation (C.V.%), and range for five salinity tolerance indices related to germination and seedling traits as comparison between control and salinity treatments.

	Control-75 mM NaCl			Control-150 mM NaCl		
	MS	R^2^	H_b_	MS	R^2^	H_b_
GTI	0.0288 **	0.67	75.68	0.0327 **	0.68	77.30
SLTI	0.0886 **	0.78	85.88	0.0683 **	0.81	88.17
SFWTI	0.0856 **	0.64	72.86	0.0682 **	0.61	69.06
SDWTI	0.2550 **	0.64	72.54	0.3020 **	0.49	48.21
WCTI	0.0046 **	0.75	83.49	0.0030 **	0.62	69.60
	Mean ± S.E.	C.V.%	Range	Mean ± S.E.	C.V.%	Range
GTI	0.90 ± 0.01	9.3	0.66–1.25	0.89 ± 0.01	9.7	0.62–1.25
SLTI	0.72 ± 0.016	15.6	0.37–1.56	0.52 ± 0.01	17.4	0.17–1.16
SFWTI	0.73 ± 0.017	20.8	0.19–1.39	0.68 ± 0.015	21.5	0.12–1.27
SDWTI	0.87 ± 0.029	26.5	0.29–2.33	1.31 ± 0.036	30.2	0.34–30.47
WCTI	0.98 ± 0.004	2.8	0.89–1.06	0.89 ± 0.003	3.5	0.69–1.01

Where, GTI is germination tolerance index, SLTI is seedling tolerance index, SFWTI is seedling fresh weight tolerance index, SDWTI is seedling dry weight tolerance index, and WCTI is water content tolerance index. ** highly significant at 0.01 probability level.

**Table 4 plants-10-02246-t004:** Identification of putative QTL associated with salinity tolerance related traits at germination and seedling stages as line main effect and line by treatment interaction.

Trait	QTL	Int. Lines	Chr.	Interval (cM)	Line Main Effect		Control		75 mM NaCl		150 mM NaCl	
					Mean	RP%	Mean	RP%	Mean	RP%	Mean	RP%
GP		Scarlett			81.5 _SC_		93.3 _SC_		73.3 _SC_		77.8 _SC_	
	QGP.S42IL.4H.a	S42IL-116	4H	1.1–40	91.9	12.7 **	95.6	2.4	97.8	33.3 **	82.2	5.7
	QGP.S42IL.6H.c	S42IL-132	6H	94.9–108.3	91.9	12.7 **	88.9	−4.8	91.1	24.2 **	95.6	22.9 **
ShL		Scarlett			7.2 _SC_		9.9 _SC_		6.2 _SC_		5.7 _SC_	
	QShl.S42IL.2H.a	S42IL-109	2H	33.9–62.7			7.7	−22.8	9.6	53.3 **	7.7	35.5 **
	QShl.S42IL.2H.c	S42IL-110	2H	89.5–97.8			9	−9.6	9.4	50.8 **	5.4	−5
RL		Scarlett			8.1 _SC_		11.3 _SC_		7.2 _SC_		5.8 _SC_	
	QRl.S42IL.2H.a	S42IL-109	2H	33.9–62.7			8.3	−26.8 *	9.67	32.6 **	7.3	26.4
	QRl.S42IL.5H.a	S42IL-126	5H	76.2–120.3			10.6	−6.6	7.22	−0.91	8.1	40.9 **
	QRl.S42IL.5H.b	S42IL-176	5H	81.3–140.1			9.4	−16.6	9.86	35.2 **	7.5	30.6 **
SL		Scarlett			15.4 _SC_		21.3 _SC_		13.5 _SC_		11.5 _SC_	
	QSl.S42IL.2H.b	S42IL-109	2H	33.9–62.7			16	−24.9	19.2	42.1 **	15	30.9 **
	QSl.S42IL.2H.d	S42IL-110	2H	89.5–97.8			19.4	−8.8	18.3	35.1 **	11.3	−1.2
	QSl.S42IL.3H.c	S42IL-115	3H	120.7–155			20.6	−3	17.4	28.4 *	9.68	−15.5
	QSl.S42IL.4H.a	S42IL-116	4H	1.1–40			19.3	−9.2	17.6	29.9 *	8.5	−25.8 *
	QSl.S42IL.6H	S42IL-132	6H	94.9–108.3			19.8	−7	14.8	9.4	15.1	31.4 **
SVI		Scarlett			1289 _SC_		1986 _SC_		992 _SC_		891 _SC_	
	QSvi.S42IL.2H.b	S42IL-109	2H	33.9–62.7			1569	−21	1537	55.0 **	1300	46.0 **
	QSvi.S42IL.4H.a	S42IL-116	4H	1.1–40			1842	−7.3	1718	73.2 **	700	−21.4
	QSvi.S42IL.6H.a	S42IL-148	6H	0.3–11.3	1004	−22 **	1356	−31.7 **	934	−5.8	723	−18.8
	QSvi.S42IL.6H.b	S42IL-132	6H	94.9–108.3	1514	17.4 *	1758	−11.5	1348	36.0 *	1438	61.5 **
SFW		Scarlett			211.5 _SC_		278.7 _SC_		184 _SC_		172 _SC_	
	QSfw.S42IL.6H	S42IL-132	6H	94.9–108.3			269.3	−3.4	196.8	7	241.9	40.6 **
SDW		Scarlett			15.4 _SC_		27		26.3 _SC_		33.4 _SC_	
	QSdw.S42IL.2H	S42IL-107	2H	12.5–41.2			21	−22.4	11.5	−56.0 **	40.6	21.8
RSR		Scarlett			1.1 _SC_		1.1 _SC_		1.2 _SC_		1.0 _SC_	
	QRsr.S42IL.5H.a	S42IL-125	5H	51.5–81.3	1.7	53.8 **	1.1	0	2.59	121.1 **	1.39	36.8
	QRsr.S42IL.5H.b	S42IL-127	5H	138.5–162.5	2.3	108 **	1.1	−7.3	1.77	51.6 **	4.1	304.6 **
WCP		Scarlett			85.5 _SC_		90.3 _SC_		85.8 _SC_		80.6 _SC_	
	QWc.S42IL.2H.a	S42IL-109	2H	33.9–62.7			87.3	−3.3	92.1	7.3 **	82.5	2.3

Where, QTL is the quantitative trail locus associated with the trait of interest, Int. lines is the introgression line that showed significant difference from Scarlett, Chr. is the chromosome name, Interval is Positions according to Comadran et al. (2012). SC is mean of Scarlett. RP% is Relative performance: RP[IL] = (LS-means [IL] − LS-means [Scarlett]) × 100/LS-means [Scarlett]. * and ** significant and highly significant at 0.05 and 0.01 probability levels, respectively.

**Table 5 plants-10-02246-t005:** Identification of putative QTL associated with salinity tolerance indices at germination and seedling stages as line main effect and line by treatment interaction.

QTL	Int. Lines	Chr.	Interval	Line Main Effect		Control-75 mM NaCl		Control-150 mM NaCl	
				Mean	RP%	Mean	RP%	Mean	RP%
Germination percentage tolerance index									
	Scarlett			0.81		0.79		0.84	
QGpti.S42IL.4H.a	S42IL-116	4H	1.1–40			1.03	30.0 **		
QGpti.S42IL.4H.b	S42IL-117	4H	17.8–49.9			1.03	30.0 **		
QGpti.S42IL.4H.c	S42IL-121	4H	51.9–81.2	1.03	24.7 *			1.09	30.7 **
QGpti.S42IL.5H.a	S42IL-126	5H	76.2–120.3	1.06	30.6 **	1.07	35.2 **		
QGpti.S42IL.5H.b	S42IL-176	5H	81.3–140.1	0.99	22.4 *	1.01	28.4 **		
QGpti.S42IL.5H.c	S42IL-127	5H	138.5–162.5	1.0	23.1 *			1.08	29.2 **
QGpti.S42IL.6H.a	S42IL-148	6H	0.3–11.3			1.10	39.4 **		
QGpti.S42IL.6H.b	S42IL-149	6H	30–51	1.01	24.7 *	1.06	34.1 **		
QGpti.S42IL.6H.c	S42IL-132	6H	94.9–108.3	1.05	29.3 **	1.03	30.0 **	1.08	28.7 **
QGpti.S42IL.7H.a	S42IL-133	7H	12.7–37.6			1.03	30.2 **		
QGpti.S42IL.7H.b	S42IL-135	7H	67.8–118.5	1.05	29.5 **			1.11	32.0 **
Seedling length tolerance index									
	Scarlett			0.58		0.64		0.54	
QSlti.S42IL.2H.a	S42IL-107	2H	12.5–41.2	0.85	46.3 **	0.98	54.2 **	0.00	
QSlti.S42IL.2H.b	S42IL-109	2H	33.9–62.7			1.23	93.9 **	0.96	78.5 **
QSlti.S42IL.2H.c	S42IL-153	2H	60.7–68.6					0.22	−59.4 **
QSlti.S42IL.2H.d	S42IL-110	2H	89.5–97.8			0.95	48.9 **		
QSlti.S42IL.3H.a	S42IL-111	3H	43.1–55.2			0.96	51.6 **		
QSlti.S42IL.3H.b	S42IL-114	3H	75.9–144.9			1.25	96.3 **		
QSlti.S42IL.5H.a	S42IL-176	5H	81.3–140.1	0.90	53.9 **	0.98	53.4 **	0.83	54.7 **
QSlti.S42IL.5H.b	S42IL-127	5H	138.5–162.5					0.22	−58.4 **
QSlti.S42IL.6H	S42IL-132	6H	94.9–108.3					0.76	41.5 **
QSlti.S42IL.7H	S42IL-139	7H	129.5–141.1					0.27	−50.0 **
Seedling fresh weight tolerance index									
	Scarlett			0.63		0.66		0.62	
QSfwti.S42IL.2H.a	S42IL-107	2H	12.5–41.2					1.00	61.8 **
QSfwti.S42IL.2H.b	S42IL-109	2H	33.9–62.7	1.1	72.7 **	1.15	73.7 **	1.06	71.7 **
Seedling dry weight tolerance index									
	Scarlett			1.1					
QSdwti.S42IL.7H.a	S42IL-133	7H	12.7–37.6	1.8	64.2 *				
QSdwti.S42IL.7H.b	S42IL-137	7H	86–127.5	1.7	59.1 *				
Water content percentage tolerance index									
	Scarlett			0.92		0.95		0.89	
QWcti.S42IL.2H.a	S42IL-108	2H	12.5–59.1			1.03	8.2 **		
QWcti.S42IL.2H.b	S42IL-109	2H	33.9–62.7	0.99	8.4 **	1.05	10.9 **		
QWcti.S42IL.2H.c	S42IL-153	2H	60.7–68.6					0.78	−12.4 **
QWcti.S42IL.3H.a	S42IL-111	3H	43.1–55.2			1.03	8.7 **		
QWcti.S42IL.3H.b	S42IL-114	3H	75.9–144.9			1.04	9.9 **		
QWcti.S42IL.4H.a	S42IL-116	4H	1.1–40			1.03	8.4 **		
QWcti.S42IL.4H.b	S42IL-146	4H	55.7–81.2			1.03	8.8 **		
QWcti.S42IL.5H	S42IL-147	5H	73.3–81.3			1.05	10.0 **		
QWcti.S42IL.6H.a	S42IL-148	6H	0.3–11.3	0.99	7.6 *	1.04	8.9 **		
QWcti.S42IL.6H.b	S42IL-129	6H	47.5–79.6			1.03	7.9 **		

Where, QTL is the quantitative trail locus associated with trait of interest, Int. lines is the introgression line that showed significant difference from Scarlett, Chr. is the chromosome name, Interval is Positions according to Comadran et al. (2012). SC is mean of Scarlett. RP% is Relative performance: RP[IL] = (LS-means [IL]–LS-means [Scarlett]) × 100/LS-means [Scarlett]. * and ** significant and highly significant at 0.05 and 0.01 probability levels, respectively.

## Data Availability

The data presented in this study are available in Appendix A.

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
