# Peer review of "Detection and Verification of QTL for Salinity Tolerance at Germination and Seedling Stages Using Wild Barley Introgression Lines"

_plants, 2021, doi:10.3390/plants10112246_

Round 1

Reviewer 1 Report

The authors have addressed my comments sufficiently.

Reviewer 2 Report

The quality of the current paper is improved. Figures and tables are easy to interpret.

This manuscript is a resubmission of an earlier submission. The following is a list of the peer review reports and author responses from that submission.

Round 1

Reviewer 1 Report

The authors used a set of wild barley introgression lines to detect the quantitative trait loci for salinity tolerance in barley. They focus on the phenotypes relating to seed germination and seedling development in response to salt treatment. Finally, they find the loci that might improve germination and seedling growth under salinity conditions. Both the idea and methods mentioned in the current paper are less of novelty. Several places in the methods are ambiguous. Why do they choose the two concentrations of NaCl?  At what stage of germination do they measure the shoot and root length?  The unit of the germination time should be "hour" instead of "day".  Units of the data in Table2 and Figure1 are missing.

Reviewer 2 Report

  1. Identification of the salinity tolerance genes at early growth stages is an important subject to food production. 
  2. In the introduction the mechanisms (as an example) should be described for surviving harsh environments apart from forming new genetic variations and alleles.
  3. The material and methods should be simplified by showing specific activities and eliminating redundant descriptions, e.g. sentence 1.

Reviewer 3 Report

This manuscript is very well written. I only have two minor comments:

  • Figures 2 and 3 are difficult to interpret. This information could be better described in a table.
  • In the "QTL detection" section in the Materials and Methods, please further describe the phenotype used as the response. Were LS means used as the phenotype?

Reviewer 4 Report

In this manuscript, the authors evaluated phenotypic traits of a barley introgression line population and a recurrent parent under salinity treatments and then performed QTL analysis for those traits. The plant materials and methods were clearly described. However, I have several concerns as follows:

First of all, there were no line numbers in the manuscript, which made detailed comments very difficult. The authors should definitely include the line numbers in a revised version.

Secondly, although the design of this experiment is ok and the materials and methods section is concise and clear enough, the section of results was not well written/organized. The authors just simply considered all traits equally important and put all the results in the main manuscript. The results section should be better organized: the most interesting results should be emphasized and less important results should be put into the supplemental. It should not simply deposit all results in the main manuscript, but you need to really carefully think about the priority of them, and make your results description more interesting for readers.

Thirdly, although the authors mentioned that some ILs were identified with good salinity tolerance performance and can be potentially used to develop new elite barley cultivars, the main objective of this study is to map QTLs for salinity tolerance traits. I cannot see from the manuscript how the identified QTLs can be used for future barley genetic studies, e.g. salinity tolerance gene cloning, or breeding practice for germplasm improvement, e.g., with marker-assisted selection or genomic prediction. I think that the usefulness of the detected QTL for future genetics/breeding studies is very important for this manuscript, and therefore, should be discussed. In addition, some salinity tolerance genes have been characterized in other crop species, e.g. GmCHX1 gene in soybean ( Nat Commun 5, 4340), which might help for figuring out the most important traits in this study and make the manuscript more readable and interesting.

Detailed comments:
The whole section of results should be better organized and more concise, especially for those paragraphs and tables for QTL identification.

All tables and figures should be better organized, and some less important results are not necessary to be displayed in the main text with a long table.

In the section of Plant material and genotyping, better include the number of lines from the BC3S4 population, although it was mentioned in the abstract.

In the section of Statistical analyses, Pearson correlation analysis was not described.

Tables 1-3: trait names should be identical among table 1-3

Figure 3: the barplots were too crowded to be visible. Should make each bar bigger and put them in supplemental. Why those traits generally had higher variation/noise than shoot length/root length shown in Figure 2?

Table 5: there are so many results to be displayed in the main text. Better select some most important results to display in the main text and put others in supplemental.